# A Descriptive Basketball Highlight Dataset for Automatic Commentary Generation

## ABSTRACT

The emergence of video captioning makes it possible to automatically generate natural language description for a given video. However, generating detailed video descriptions that incorporate domain-specific information remains an unsolved challenge, holding significant research and application value, particularly in domains such as sports commentary generation. Moreover, sports event commentary goes beyond being a mere game report, it involves entertaining, metaphorical, and emotional descriptions. To promote the field of sports commentary automatic generation, in this paper, we introduce a novel dataset, the Basketball Highlight Commentary (BH-Commentary), comprising approximately 4K basketball highlight videos with groundtruth commentaries from professional commentators. In addition, we propose an end-to-end framework as a benchmark for basketball highlight commentary generation task, in which a lightweight and effective prompt strategy is designed to enhance alignment fusion among visual and textual features. Extensive experiments on the BH-Commentary dataset demonstrate the validity of the dataset and the effectiveness of the proposed benchmark for sports highlight commentary generation. (The dataset is available at https://anonymous.4open.science/r/dataset-DC8E)

## CCS CONCEPTS

• **Computing methodologies** → **Computer vision tasks**.

## KEYWORDS

Dataset, Video Captioning, Basketball Commentary Generation, Vision-Language

**ACM Reference Format:**
. 2018. A Descriptive Basketball Highlight Dataset for Automatic Commentary Generation. In *Proceedings of Make sure to enter the correct conference title from your rights confirmation emai (Conference acronym 'XX)*. ACM, New York, NY, USA, 10 pages. https://doi.org/XXXXXXX.XXXXXXX

## 1 INTRODUCTION

Video captioning [48, 63] stands as a challenging and essential task in both the computer vision and natural language processing communities. Aimed at automatically generate the description about the visual content of a given video in natural language, this task has gained significant attention in recent years due to its importance across various applications. One good example is sports

commentary generation (especially for team sports such as football, basketball, and volleyball etc). Figure 1 illustrates the distinction between the conventional video captioning task and our sports commentary generation. We can notice that the conventional video captioning can solely provide a macroscopic perspective description of the video (e.g., a scene featuring players are playing basketball). In contrast, the commentary generation is capable of offering more vivid description of individual technical movements and the coordination between team members (e.g., Player A launches a long pass from the backcourt, setting up Player B for a one-handed slam dunk, showcasing flawless teamwork.).

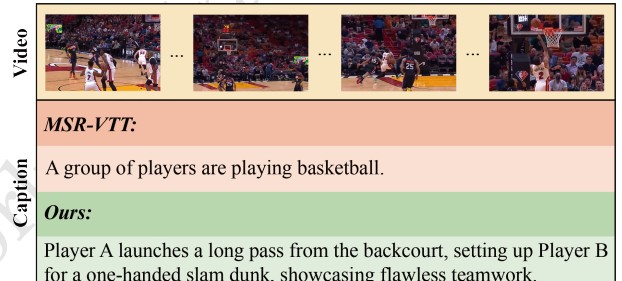

**Video**

**MSR-VTT:**
A group of players are playing basketball.

***Ours:***
Player A launches a long pass from the backcourt, setting up Player B for a one-handed slam dunk, showcasing flawless teamwork.

**Caption**

**Figure 1: Comparison between previous work and ours. Our commentary generation presents a more realistic and vivid scene**

For a sports highlight video that showcases exquisite individual move and teamwork, the key of commentary generation lies in capturing the the visual characteristics of athletes' technical movements and map them into statements that contain technical terms and descriptive words. However, for basketball highlight video, this presents several challenges. Firstly, the basketball highlights usually contain players' gorgeous technical movements and exquisite teamwork, which provides a visual basis for downstream generative model. Accurately and effectively capturing and representing these visual features from video can offer more informative cues for commentary generation. Several recent studies [50, 64, 70, 73] focused to leverage the action information contained in videos to enhance the downstream task. However, such studies usually employ multiple feature extractors that trained for visual understanding tasks to extract 2D and 3D visual features. Although these approaches have shown promising results, there raises problems about the extent to which these extracted features from off-line extractors can be effectively adapted to suit the requirements of the captioning task. Secondly, the generated commentaries should highlight the player's gorgeous moves and coordination, that is the model should be able to generate commentaries containing the player's technical movements based on visual information. Most of the existing research [3, 19, 47, 55] tends to emphasize the effective exploitation of visual features, with few taking into account the significance of cross-modal interactive fusion and explicitly leveraging such

interaction to enhance the downstream generation. Moreover, as for basketball highlight video, the commentary on it must not be a simple description of the player's actions. While it is difficult to train a model that can generate commentary that is descriptive and professional in content on the basis of existing video captioning datasets. In this context, there is a significant need for a readily accessible sports commentary dataset, annotated by professional sports analysts, to facilitate research in this area.

To explicitly tackle these challenges and develop a practical sports highlight commentary generation system, particularly in the context of basketball highlight video, in this paper, we propose an end-to-end framework for basketball highlight commentary generation. Specifically, for visual feature extraction, taking inspiration from the recent works of transformer-based models in video understanding [2, 13, 14], we utilize a video transformer to extract features with the original video as input. In contrast to employing separate offline 2D and 3D visual extractors, our model integrates visual extraction within a unified framework, along with subsequent multi-modal feature encoding and commentary generation modules. This integration enhances the suitability of the extracted visual features for downstream tasks. As for multi-modal feature encoding, a lightweight but effective prompt strategy is designed to promote the interaction fusion between visual and textual features, which prompts the model to focus on the visual representations that are most relevant to the text. It is worth noting that in our proposed model, each component is integrated into the unified framework, which makes the components of the model compatible and mutually reinforcing. Moreover, to initiate shareable research in this emerging field, we are introducing a new dataset, called Basketball Highlight Commentary (BH-Commentary). This dataset comprises 4,396 high-definition NBA basketball highlight videos from the Tencent Video website, each of which is annotated with detailed descriptive commentary.

In summary, the main contributions of this work are summarized as follows:

- We collect a novel high-quality dataset for sports highlight commentary generation, which contains *4K* basketball highlight videos from websites and corresponding commentaries from professional commentators.
- We propose an end-to-end benchmark model for sports highlight commentary generation, which integrates visual feature extraction, multi-modal feature encoding and commentary generation task into a unified framework.
- A lightweight and effective prompt strategy is designed to promote multi-modal feature interactive fusion.
- Extensive experiments on the collected dataset demonstrate the proposed benchmark model's effectiveness and the validity of the dataset.

## 2 RELATED WORKS

### 2.1 Video Captioning

Video captioning aims to generate a condensed natural linguistic sentence that describes the main event of a video. Early researches adopt the template-based strategy to generate video captions [24, 60], this sort of methods usually align the sentence components to the detected visual content, and generate the description

based on the pre-defined templates, which are typically limited by the fixed templates. Recent works usually adopt encoder-decoder structure for this task [17, 48, 68], where the encoder translates the input video to visual features, and the decoder integrates the encoded visual features and generate a natural sentence. Since without bounded by the pre-defined template, such methods can generate captions with more flexible sentence patterns. Specifically, based on the extracted visual or visual-linguistic feature, [47, 66, 69] utilize the LSTM/GRU-based architecture for caption generation task, [50, 63, 72] use transformer-based model for video captioning generation. Unlike the above models that adopt offline feature extraction, we take an end-to-end approach to integrating feature extraction with downstream task.

### 2.2 Visual Extractor

Transformer [53], adopting an attention-based encoder-decoder structure, has demonstrated promising performance on the NLP tasks. Inspired by the outstanding ability on sequence modeling, some recent researches explore transformer-based structure in the field of computer vision, achieving remarkable results on basic CV tasks [13, 18, 58, 65]. Since the competitive modeling capabilities, the visual transformers have achieved impressive performance improvement compared with the traditional methods. The application of visual transformer to video field is also gaining increasing attention. In order to cope with the characteristics of videos with long sequences, Neimark *et al.* [40] adopt temporal attention-based encoder, which could attend to all tokens in the input sequence, making the model capable of handling long sequences. Arnab *et al.* [2] introduce a transformer-based model to employ spatial-temporal attention for better video representation. Zhang *et al.* [67] propose stacked attention to aggregate spatio-temporal information contained in the video for improving representation learning. Moreover, inspired by the success of Swin Transformer in image domain [35], Liu *et al.* [36] further propose Video Swin Transformer, which introduces an inductive bias of locality in spatiotemporal domain into transformer structure, obtaining promising video representation.

### 2.3 Vision Language Model

Joint vision language understanding associates the computer vision and natural language processing together, and has attracted increasing attention from the two fields. Recent researches [27, 50] have shown the success in the field of multi-modal representation learning for vision-language understanding and generation, including downstream tasks like video-language retrieval [6, 16], video question answering [26, 32], and video captioning [56, 62]. In order to get better performance, most language models tend to adopt large scale training data, causing the loss of computation and memory. With the success of the language pre-training and video-text pre-training strategy, recent works attempt to employ the pre-trained language models to the vision language task. For example, by freezing the weights of a pre-trained language model, [1, 52, 57] show promising results in vision language tasks. Moreover, masked language models also show success in language works [12, 25, 31], which pre-trains a transformer-based structure to learn language representations, achieving competitive performance in downstream tasks after being fine-tuned. The success of masked

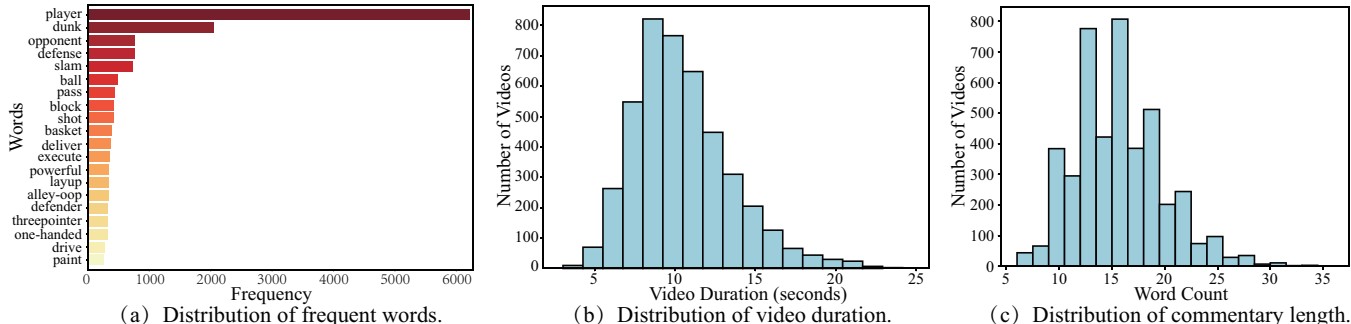

(a) Distribution of frequent words.    (b) Distribution of video duration.    (c) Distribution of commentary length.

**Figure 2: Illustrations of BH-Commentary dataset statistics. (a-c) Distribution of frequent words, video duration and commentary length of the dataset in English version.**

language models also drives the exploration the works of applying it to the multi-modal representation model with paired visual-textual data [15, 29, 37], which show competitive performance on vision language tasks.

| Dataset | Domain | #Video | #Sentence | Total Dur(h) |
|---|---|---|---|---|
| MSVD [9] | Open | 1970 | 70k | 5.3 |
| MSR-VTT [59] | Open | 10k | 200k | 41.2 |
| MPII-MD [46] | Movie | 68.3k | 68.3k | 73 |
| TACoS [45] | Cooking | 127 | 11.8k | - |
| YouCook [11] | Cooking | 88 | 2.7k | 2.3 |
| YouCook2 [71] | Cooking | 2k | 15.4k | 176 |
| ActivityNet-Caption [7] | Open | 20k | 100k | 840 |
| SoccerNet-caption [39] | Soccer Game | 0.9k | 36k | 715.9 |
| SVCDV [44] | Volleyball | 4.8k | 44k | - |
| FSN [64] | Basketball | 2k | 6.5k | - |
| **BH-Commentary** | Basketball | 4.3k | 4.3k | 10.1 |

**Table 1: Comparison existing video captioning datasets.**

## 3 BASKETBALL HIGHLIGHT COMMENTARY DATASET

Basketball Highlight Commentary (BH-Commentary) is a basketball highlight video commentary generation dataset. Each highlight clip is annotated with an description of its content. Unlike previous video captioning datasets that describe motions from a macro perspective, this dataset focuses on providing a lively language of commentary on the technical movements of players in basketball videos, where each comment corresponds to one event highlight. In the following, we introduce the dataset collection process and provide a comprehensive statistical analysis on this dataset.

### 3.1 Dataset Collection

We collect 4,800 highlight videos from the NBA's 2020-2023 season from websites. And we filter out videos that are too short and had poor visual quality, ultimately choosing 4,396 videos with diverse and detailed motions for the final annotation process. Basketball highlights videos in our dataset involve six categories of basketball actions, including pass, dunk, block, shot, steal and layup, as shown in Figure 3(a). All videos are available at 25fps in two resolutions:

480p and 720p. The commentaries from professional commentators are initially presented in Chinese version in audio form, which we transcribe into English text through transcription and proofreading. Moreover, in line with conventional captioning datasets, we offer the anonymized version of the captions in which specific players' names are replaced with generic tokens. In fact, most of existing captioning models are not capable to accurately identify the individuals featured in the videos. Since generating accurate names would be nearly impossible without the inclusion of specifically designed modules for identity classification and identification.

| Dataset | verb per sent | noun per sent | adj per sent | adv per sent |
|---|---|---|---|---|
| MSR-VTT [59] | 1.84 | 3.20 | 0.60 | 0.15 |
| BH-Commentary | **1.42** | **6.31** | **1.30** | **0.55** |

**Table 2: Comparison of the average number of verbs, nouns, adjectives and adverbs per sentence of the our dataset and MSR-VTT dataset.**

### 3.2 Dataset Statistics

Our dataset includes 4,396 videos, each of which corresponds to one annotated statements from professional commentators. Each video has an average of 15.3 words. On average, each word describes 0.5s in video and 4.8% of the entire video, which demonstrates that our annotations are informative and detailed, comprehensively encompassing the contents in the video. Table 1 provides a comparison of major statistics between our dataset and other existing popular video captioning datasets. Unlike other datasets collect videos from common domain or generate them virtually that stand out with longer total video duration, our dataset mainly focuses on highlight from real basketball game scenes. Based on a limited number of basketball game highlight and the corresponding commentaries from professional commentators, our dataset contains 4,396 highlight videos with a total of 10.1 hours and the same number of annotations. As shown in Figure 2(a), in our dataset, the most frequently occurring words are the names of the players, followed by words that are semantically related to basketball and associated elements. In addition, the distribution of video duration of our dataset is shown in Figure 2(b), the longest video lasts 27.8s and

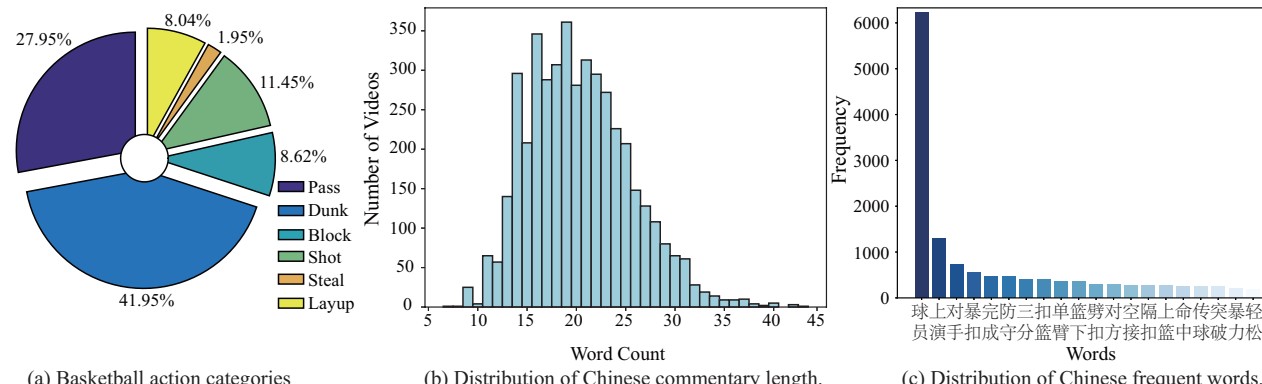

(a) Basketball action categories     (b) Distribution of Chinese commentary length.     (c) Distribution of Chinese frequent words.

Figure 3: Illustrations of BH-Commentary dataset statistics. (a) Distribution of basketball action categories.(b-c) Distribution of frequent words and commentary length in Chinese version.

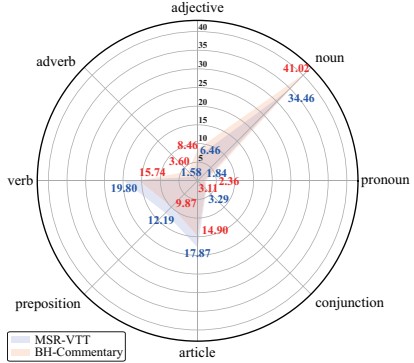

Figure 4: The parts of speech distribution of BH-Commentary and MSR-VTT dataset. All the values in the figure are the percentage of parts of speech ratio. There are more adjectives and adverbs in BH-Commentary, as this is commentary generation dataset focusing on providing descriptive language for players' motions.

the shortest one lasts 3.1s, and the average video duration is 10.5s. And the distribution of commentary length of our dataset is shown in Figure 2(c), the length of the commentaries in our dataset varies, with the longest being 36 words and the shortest being 6 words.

As for the Chinese version of our dataset, Figure 3(b-c) illustrate the distribution of frequent words and commentary length in Chinese version. As shown in Figure 3(b), the length of the Chinese commentaries in our dataset varies, with the longest being 43 words and the shortest being 7 words. In addition, similar to the English version, in Chinese version of our dataset, the most frequently occurring words are the names of the players, followed by terms that are semantically related to basketball and associated elements, as shown in Figure 3(c).

Moreover, we conduct a parts of speech analysis on our dataset in comparison with MSR-VTT [59]. As depicted in Figure 4, our dataset exhibits a higher proportion of nouns, adverbs, and adjectives, which underscores our dataset's increased focus on players,

their technical movements, and the accompanying descriptive elements. In Table 2, the comparison of the average number of verbs, nouns, adjectives, and adverbs per sentence further demonstrate the descriptive advantage of our annotations. In each highlights video, our annotations feature a higher count of descriptive words per sentence, this is in line with our objective: delivering vivid commentaries for sports highlights. And for dataset splitting, we take the same settings as MSR-VTT dataset that we randomly divided the dataset into training, validation, and testing sets with proportions of 65%, 5%, and 30%, respectively.

## 3.3 Novelty

Committed to advancing the researches about video captioning/ description generation, multiple datasets covering various domains have been introduced. In general, video captioning tasks can be primarily categorized into two families: single event caption generation [9, 59] and multiple events caption generation [39, 64, 71]. As shown in Table 1, due to the objective reasons such as the difficulty of video collection and annotation, previous studies rarely focus on sports video description generation. Some studies, such as [39], utilize virtual methods such as games to create sports game videos for building dataset. [44] and [64] built datasets of video descriptions generation based on volleyball and basketball games. However, all the previous works on sports video description generation focus on the relaying of players' movements during a game. In this work, based on the basketball games, we propose the fist dataset focusing on sports highlight commentary generation, which provides more descriptive language than mere statements of action. Compared with the existing basketball video description dataset FSN [71], our dataset tends to provide emotional and descriptive commentary for basketball highlights, which makes it easier for the audience to empathize with the excitement of the game.

## 4 COMMENTARY GENERATION MODEL

The goal of sports highlight commentary generation is to automatically generate eloquent and descriptive sentence to paint a vivid picture of the technical movements executed by the players in the video. This challenge raises the question of how to enable the

**Figure 5: The overall architecture of the proposed model. We formulate the commentary generation as a sequence-to-sequence task, the raw video frames and the text are first encoded by the visual and textual encoder, respectively. The prompt embedding that utilized to facilitate the multi-modal feature fusion is aggregated with the input and the previous state, and is concatenated with the multi-modal embeddings, which are further input to the multi-modal encoder. Then the language decoder head autoregressively generates the output commentary based on the multi-modal representaions.**

efficient mapping of visual input to commentary output. Furthermore, the well-extracted visual features could serve as crucial visual cues for commentary generation. To tackle these problems, we first introduce a unified end-to-end multi-modal encoding framework, treating the automatic generation of sports highlights commentary as a sequence-to-sequence task, as explained in Section 4.1. Then, an effective prompt strategy is devised to enhance the alignment of multi-modal representations in Section 4.2. And the strategy of training and inference are introduced in Section 4.3.

## 4.1 Model Architecture

We wish to design an architecture that can effectively map the sports highlight video to corresponding descriptive commentary. To achieve this goal, we introduce an end-to-end framework that takes raw sports highlight video frames as input and generates natural language commentary for input content description. Figure 5 shows the overview of our proposed benchmark model. In detail, given a pair of video $\{f_t\}|_{t=1}^{T}$ and text sequence $\{s_n\}|_{n=1}^{L}$, where $T$ represents the number of sampled frames from the input video, and $L$ denotes the length of the sentence. We first separately encode them using individual encoders to obtain unimodal features, the visual encoder extracts visual features from the raw sports highlight video frames, while the text encoder embeds the textual representation. Subsequently, the multi-modal encoder further encodes the multi-modal representation based on both the visual and textual features. And the commentary is generated in an auto-regressive manner. The detailed description of each module are given as follows.

**Visual Encoder.** Drawing inspiration from the success of various transformer-based model for video representation learning in long-range temporal relationship modeling [5, 36, 61], recent advancements in video-language research [31, 51] have begun to leverage the success of video transformers, showcasing improved performance in downstream tasks. In this paper, we employ the Video Swin Transformer [36] (VST) as visual backbone for visual feature extraction, based on the frames from raw input video.

The visual encoder takes a sequence of $T$ frames $f \in \mathbb{R}^{H \times W \times 3}$ sampled from raw video as input, where $H$ and $W$ refer to the height and width of each frame. Then the grid features are extracted from the last encoder block of VST, resulting in grid features with dimension of $\frac{T}{2} \times \frac{H}{32} \times \frac{W}{32} \times 8C$, where $C$ represents the channel dimension. These grid features are then tokenized along the channel dimension, yielding a total of $\frac{T}{2} \times \frac{H}{32} \times \frac{W}{32}$ video tokens, with each token being an $8C$-dimensional feature vector. For more in-depth information, please refer to [36]. These extracted visual features are then utilized as input for the multi-modal fusion encoder to facilitate the learning of cross-modal representation.

**Textual Encoder.** For text encoding, the input text sentence is first tokenized into a sequence of $N$ tokens $\{t_n\}|_{n=1}^{N}$. And two special tokens, [CLS] and [SEP], are inserted at the start and the end of the token sequence. Then, like previous works [27, 37], we utilize a lightweight word embedding layer [21] to obtain textual embeddings, which are concatenated with the visual features and then input to the multi-modal encoder.

**Multi-modal Encoder.** We utilize a transformer-based multi-modal encoder for multi-modal features encoding. Specifically, the multi-modal encoder takes two modal inputs, which correspond to the

visual and textual features extracted from the two unimodal encoders. Denoting the encoded visual and textual embeddings as $E_v \in \mathbb{R}^{T \times k}$ and $E_t \in \mathbb{R}^{N \times k}$, we then concatenate these two embeddings as input to the multi-modal encoder, denoted as $E_m = [E_v; E_t] \in \mathbb{R}^{(T+N) \times k}$, where $[;]$ denotes concatenation and $k$ denotes the dimension of hidden state. To obtain the cross-modal representations, the visual and textual embeddings are combined through cross-attention operations. Then we conduct sequence to sequence generation process to implement commentary generation, where we employ a causal self-attention mask, ensuring that a caption token can only attend to the previously generated output tokens.

Through our generic design, we enable end-to-end training for commentary generation using the raw video frames. Furthermore, by leveraging the versatility of the transformer architecture, our model can handle video sequences with variable lengths.

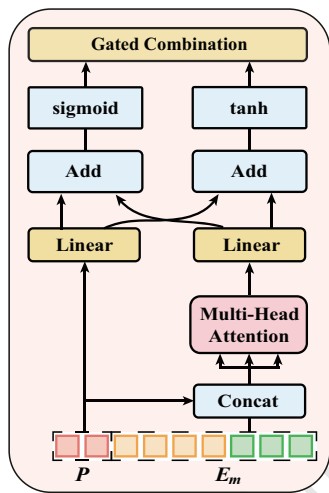

**Figure 6: The illustration of prompt embedding updating scheme. The prompt embedding is selectively updated according to the input using the multi-head attention with a residual connection.**

## 4.2 Multi-modal Fusion via Prompt

Recently, a line of works show promising performance in obtaining the desired output through prompt designing [20, 28, 30, 41]. Instead of designing manually, soft prompt is proposed as a series of continuous embeddings that are prepended to the input and updated throughout training. In this work, we propose to utilize a lightweight soft prompt strategy using the attention network with a residual connection for promoting multi-modal features interaction and fusion. Instead of using the original prompt setting [28], we pass the soft prompt embedding and multi-modal embeddings through the attention network with a residual connection. Subsequently, we reparameterize the prompt and prepend it to the multi-modal embeddings before feeding it into the multi-modal encoder. In specific, as shown in Figure 6, we set a sequence of soft prompt embedding $P = [p_1, \ldots, p_n] \in \mathbb{R}^{n \times k}$, here $n$ and $k$ denote the number of and the dimension of prompt vectors respectively.

With multi-head attention, we can aggregate the feature from both the multi-modal embeddings $E_m$ and the prompt embedding $P$, and the prompt embedding can then be selectively updated according to both the current input and the previous state with residual connection. Then the prompt-concatenated multi-modal representations are further encoded through the multi-modal encoder. The entire process above is denoted as below:

$$
\begin{aligned}
A &= \text{FFN}(MHAtt([P, E_m])), \\
S &= \tanh(W_{sp}\, P + W_{sa}\, A + b_s), \\
Z &= \text{sigmoid}(W_{zp}\, P + W_{za}\, A + b_z), \\
P' &= (1 - Z) \odot S + Z \odot P,
\end{aligned}
\tag{1}
$$

where FFN denotes feed-forward network, $MHAtt$ denotes multi-head attention in transformer network [53], tanh and sigmoid denote activation functions, $\odot$ denotes Hadamard product, $W_{sp}$, $W_{sa}$, $W_{zp}$ and $W_{za}$ are trainable weights, $b_s$ and $b_z$ are trainable bias.

## 4.3 Training

***Train Setting.*** The visual encoder is pre-trained on the Kinetics action recognition task [8]. During training, the model takes video and text input, which are further input to the visual and textual encoder for feature extraction. The prompt embedding is jointly updated with the model during training. Furthermore, all textual tokens have complete attention not only to the visual tokens but also to the prompt, ensuring that the prompt can enhance the comprehensive interaction of multi-modal features, which allows the model to effectively utilize both visual and textual modalities to generation accurate and descriptive commentary.

***Inference.*** During inference, the model solely takes the video as input, and the commentary is generated in an auto-regressive manner. The model generates one textual token at a time, using the tokens generated thus far as inputs for the multi-modal transformer encoder. And the prompt is no longer updated, instead, it serves the purpose of facilitating the commentary generation.

## 5 EXPERIMENT

In this section, we demonstrate the effectiveness of our benchmark model on its ability of generating sports highlight commentary. We conduct experiments on BH-Commentary dataset, which is specifically built for this task, and we compare our model to the state of the art. We first introduce the experimental setting in Section 5.1, and the ablation study is conducted in Section 5.2. Finally, we present the experimental results and analysis in Section 5.3.

## 5.1 Experimental Setting

***Metrics.*** We adopt several widely-used evaluation metrics, including BLEU@4 [42], METEOR [4], Rouge-L [33], and CIDEr [54] to measure the performance of the benchmark model. We calculate these metrics using the standard COCO evaluation tools [1] [10].

***Implementation Details.*** Our model is implemented in PyTorch [43], the visual encoder is initialized with Kinetics-600 pre-trained weights, the textual encoder is initialized from pre-trained BERT-Base [12], and the multi-modal encoder is initialized randomly. The number of prompt vectors is set to 8, which is equal to 1% of the

---

[1] https://github.com/tylin/coco-caption

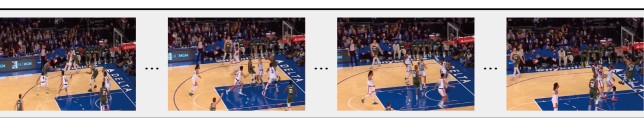

**Groundtruth:** (a)

Player A flicks a clever pass, helping Player B rise for a sky-high dunk.

**Baseline:**

Player a's clever pass, and Player b for a one-handed slam dunk.

**Ours:**

Player a's clever pass sets up Player b for a powerful dunk.

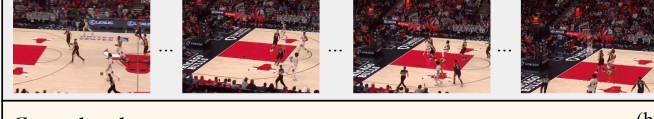

**Groundtruth:** (b)

Player A initiates a fast break, throws an alley-oop, and Player B takes flight, performing a two-handed dunk.

**Baseline:**

Player a's clever pass Player b, and Player b for a one-handed slam dunk.

**Ours:**

Player a's precise pass sets up player b for a two-handed dunk.

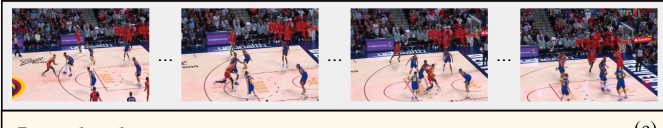

**Groundtruth:** (c)

Player A drives straight in, showcasing what "unstoppable" means with a sky-high slam dunk.

**Baseline:**

Player a's clever pass, Player a executes a one-handed slam dunk.

**Ours:**

Player a dribbles past the defense, executing a one-handed slam dunk.

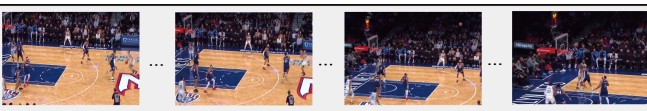

**Groundtruth:** (d)

Player A takes two steps beyond the three-point line, releases a shot over the defender, and swishes it through the net with a beautiful arc.

**Baseline:**

Player a hits a long-range three-pointer, hits a three-pointer.

**Ours:**

Player a nails a long-range three-pointer, showcasing a long-range three-pointer.

**Figure 7: Qualitative examples generated by our benchmark model.**

number of multi-modal representation vectors, meeting the need for lightweight. For multi-modal encoding, we adopt the transformer-based structure with 12 layers and 768 dimensional hidden states. The whole model is trained in end-to-end manner. In addition, we adopt the AdamW [22] optimizer with an initial learning rate of 3e-5 and use a learning rate warm-up during the early 10% training steps followed by linear decay.

## 5.2 Ablation Study

To verify the effectiveness of the designed prompt, we show the performance changes in the last block of Table 3 by removing the prompt and simply input the concatenated multi-modal embeddings to the multi-modal encoder for the following commentary generation, which obviously results in a decline in model performance across all evaluation metrics. The results suggest that the performance of commentary generation can be greatly lifted by using prompt for promoting multi-modal feature fusion. Moreover, the model without prompt setting is selected as the baseline model for comparison in experiment analysis, which is discussed in the following section.

## 5.3 Results and Analysis

***Compare to the State of the Art.*** We consider four up-to-date baselines for comparison. Table 3 lists the main results on the commentary generation task. According to Table 3, our benchmark model demonstrates the capability to generate more accurate and higher-quality commentaries when compared with the baselines. We attribute the superior performance of our benchmark model to two main factors. Firstly, the end-to-end setting enables the modules to be iteratively updated within a unified framework, enhancing the compatibility between each module. Additionally, the

| Model | Bleu@4 | METEOR | Rouge-L | CIDEr |
|---|---|---|---|---|
| Swinbert [34] | 3.2 | 12.5 | 27.6 | 11.7 |
| UniVL [38] | 2.9 | 11.3 | 18.2 | 6.3 |
| UniVL+MELTR [23] | 3.6 | 12.4 | 27.8 | 11.4 |
| CoCap [49] | 3.8 | 12.6 | 27.7 | 11.8 |
| w/o prompt | 3.2 | 12.4 | 27.3 | 10.9 |
| ours | **4.1** | **12.9** | **28.7** | **12.2** |

**Table 3: Comparison of the proposed benchmark model with the state of the art works for commentary generation task on BH-Commentary dataset.**

well-designed prompt plays a crucial role in facilitating the fusion of multi-modal features, thereby promoting the downstream commentary generation task. However, from an intuitive perspective, the models' performance metrics on our dataset seem comparatively lower than on other datasets. This can be attributed to the inherent complexity of generating commentary for sports highlight videos. The intricate sentence structures and highly descriptive content in our dataset pose significant challenges to the learning process of the model. Despite this, our model serves as an inspiration for future efforts to address the challenges in sports highlight commentary generation.

***Qualitative Analysis.*** To further qualitatively assess the performance of our benchmark model in the commentary generation task, Figure 7 shows several examples about the highlight videos and corresponding commentaries obtained from groundtruth, baseline model and benchmark model. These examples indicate that out benchmark model can recognize the visual contents, and generate accurate terms and descriptive sentences. In both examples, the

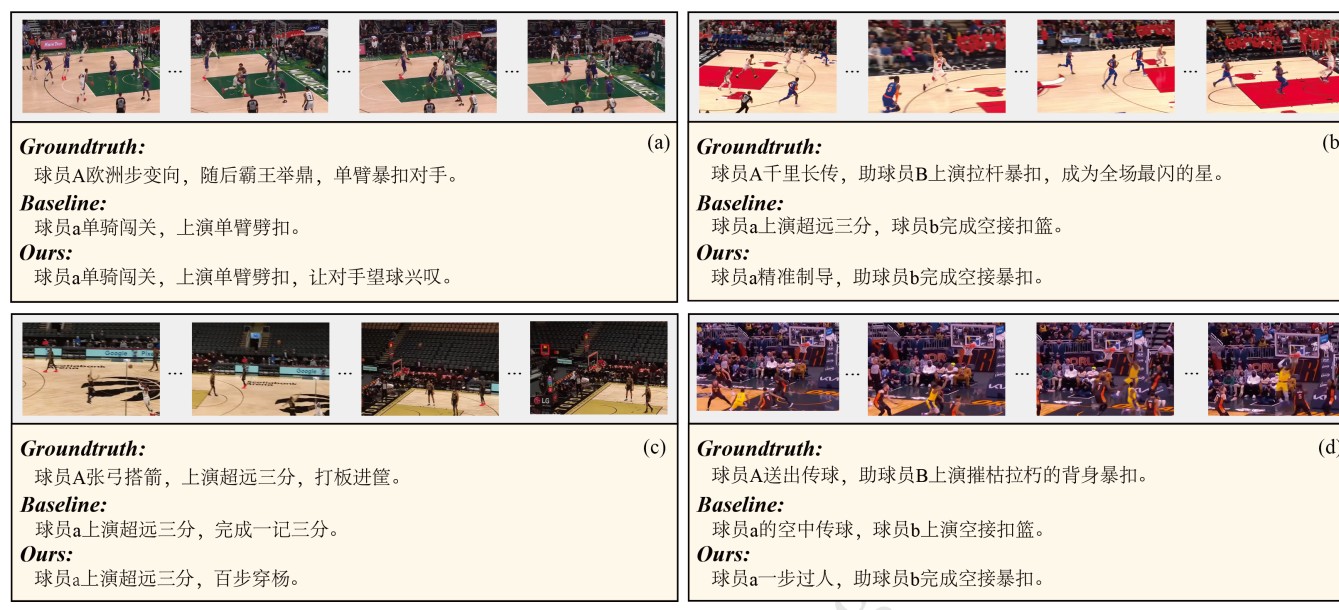

**Figure 8: Qualitative examples generated by our benchmark model in Chinese version.**

generated commentaries can cover the key technical moves of the players. In specific, for the first example, the video showcases the clever pass and powerful dunk executed by players. As shown in Figure 7(a), our model can accurately generate commentary that encompasses detailed actions and matches the groundtruth. In contrast, the baseline model is not able to generate the appropriate vocabulary to link the two actions, thereby failing to depict the coordination between the players. For the second example shown in Figure 7(b), we can observe that our model accurately generates the basketball skill action "two-handed dunk," which aligns with the groundtruth. In contrast, the baseline model fails to do so. As shown in Figure 7(c), in the third example, the commentary produced by our model accurately describes the actions "dribbles past the defense" and "one-handed slam dunk," which correspond to "drives straight in" and "sky-high slam dunk" in the groundtruth, respectively. While the baseline model, though correctly generating the term "one-handed slam dunk," provides inaccurate information with "clever pass." In the fourth example, as shown in Figure 7(d), our model delivers precise commentary, stating that the player successfully nails a long-range three-pointer, followed by a concise summary description. While the baseline model just repeats the given information.

***Chinese Version.*** We also conducted a qualitative analysis on the Chinese version of the dataset, as illustrated in Figure 8. Our model accurately provides descriptions in the commentary, enriched with the suitable idioms. Specifically, in the first example, the highlight video showcases the player's dynamic and powerful dunk as he breaks through the defense. As shown in Figure 8(a), our model excels in generating accurate commentary that matches with the groundtruth and provides a summary description. In the second example, shown in Figure 8(b), we can observe that our model accurately generates the basketball skill actions based on precise pass and dunk, which is consistent with the groundtruth. In contrast,

the baseline model produces inaccurate information with "three-pointer." As shown in Figure 8(c), in the third example, the commentary generated by our model accurately provides the term "three-pointer" and corresponding description. While the baseline model just repeats the given information. In the fourth example shown in Figure 8(d), our model outperforms the baseline by offering additional detailed information about Player A before executing the pass, which is not mentioned but actually true in the video.

***Shortcoming.*** During experiments, we also find some shortcomings in our benchmark model. Our model may sometimes fail to recognize similar movements, such as layups and dunks. Moreover, as shown in Figure 7 and Figure 8, compared with the groundtruth, out generated commentary may lack some background information description like "fast break", and does not have the ability to generate such statements like "showcasing what "unstoppable" means with a sky-high slam dunk". Our benchmark model serves as inspiration here. Addressing the challenge of enabling the model to understand these statements and generate them in the appropriate context remains a task that needs further exploration in subsequent research.

## 6 CONCLUSION

In this work, we create a descriptive basketball highlight video dataset for sports highlight commentary generation task. We propose a benchmark model for this task and outperform the state of the art models. Extensive experiments demonstrate the effectiveness of the proposed benchmark model and validity of the collected dataset. Due to the rich content of descriptive commentary, it is apparent that there is room for improvement in the performance of the model for the sports highlight commentary generation task, which remains research direction for subsequent studies. Our benchmark model serves as an inspiration here for the further research.

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
