# OpenReview forum: "A Descriptive Basketball Highlight Dataset for Automatic Commentary Generation"
_acmmm.org/ACMMM/2024/Conference — MM2024 Poster_

### Official Review · Reviewer_XYBL · 2024-05-20

**Rating:** 3
**Confidence:** 3

**Summary:**

This study introduces a novel dataset aimed at automatic sports commentary generation. It includes around 4,000 basketball highlight videos, each paired with professional ground-truth commentaries. Furthermore, an end-to-end framework has been developed for this purpose, employing a lightweight and efficient prompt strategy. Experimental results validate the proposed dataset and confirm the effectiveness of the framework.

**Strengths:**

1. A high-quality dataset for generating sports commentaries is introduced.
2. The proposed model outperforms current state-of-the-art methods across various metrics.
3. The writing and organization are well-executed.

**Limitations:**

1. The dataset's diversity is limited because it includes only one type of sport, which means its generalization ability is not guaranteed.
2. The authors claim that the proposed dataset provides more emotional and descriptive commentary compared to FSN [71]. More statistics and examples are needed to validate this claim.
3. While qualitative results of the baseline model are provided, results from existing methods are also necessary to demonstrate the superiority of the proposed method.
4. The performance of the proposed model on existing datasets is not discussed.
5. There is no failure case study or discussion of the model's limitations.

**Suitability:**

3

---

### Official Review · Reviewer_ZWD4 · 2024-05-24

**Rating:** 2
**Confidence:** 3

**Summary:**

This paper proposes a BH-Commentary dataset for sports highlight commentary generation and proposes a model, that integrates visual feature extraction, multi-model feature encoding, and commentary generation task by a unified framework. The effectiveness of the proposed model is verified through experimental evaluation of the BH-Commentary dataset.

**Strengths:**

The strengths of this paper include:
1. This paper proposes a descriptive basketball highlight dataset for automatic commentary generation.
2. This paper builds a benchmark model.
3. This paper analyzes the experimental results.

**Limitations:**

Review and Evaluation of Weaknesses:
1. "Anonymous Authors" seems to have been deleted in the paper template.
2. The validation analysis of the proposed model is insufficient. Table 3 only compared with four baselines and one ablation method.
3. The figures of the paper are confusing. The identifiers in Figure 5 use p_1, v_1, t_1 and are not consistent with the writing method in the text (video: f, text: s), and the expression is unclear.
4. Figure 7 & Figure 8 Except for the language version, it seems that no more additional information is introduced, which is redundant.

**Suitability:**

3

---

### Official Review · Reviewer_gbmz · 2024-05-25

**Rating:** 4
**Confidence:** 2

**Summary:**

The paper proposes the Basketball Highlight Commentary (BH-Commentary) dataset to faciliate research of automatic commentary generation. For building the dataset, the authors have collected approximately 4K basketball highlight videos from websites and corresponding ground-truth commentaries from professional commentators. In addition, the paper propose a benchmark model for  commentary generation from sport videos. Extensive experiments on the BH-Commentary dataset show the effectiveness of the proposed benchmark model, and reveal the challenges of sports highlight commentary generation.

The proposed dataset brings new challenges comparing to existing Video Caption datasets. From both academic and industrial perspective, this work potentialy opens interesting research directions for sports highlight commentary generation. However, the paper lacks details of some components of the benchmark model. The missing details reduce the validity and reproducibility of the experiments performed. Therefore, the paper is rated as borderline accept. See also the details in the following Strengths and Limitations sections.

**Strengths:**

- The paper has constructed a novel dataset for automatic commentary generation. The dataset brings new challenges that need to generate emotional and descriptive commentary compared to the existing video caption dataset such as MSR-VTT and FSN. The details of the data collection process and data statistics are well described in the paper.

- The paper has conducted a benchmark on the proposed dataset, and clarified the shortcomings of the baseline models through the error analysis. Since the baseline methods are hard to generate the emotional and descriptive commentary in the proposed dataset, there is room for improvement in the performance of the model for the sports highlight commentary generation task, which opens the new research direction for subsequent studies.

**Limitations:**

- Missing details of some components in the proposed benchmark model.
The benchmark model includes a prompt embedding updating scheme and Language Decoder Head. For the prompt embedding updating scheme, Figure 5 and 6 look inconsistent. Figure 6 in the paper shows that the prompt embedding updating scheme takes learnable prompt $P$ and multi-modal feature $E_m$ to output $P^{\prime}$ while $P^{\prime}$ is generated from only $P$ input in Fig. 5.  The paper obscures the end-to-end learning pipeline with prompt embedding updates. Also, there are no details of Language Decoder Head in the paper.

- The same evaluation metrics are used for commentary generation as for conventional Video Captioning.
The paper mentions that the propsoed dataset tends to provide emotional and descriptieve commentary as the ground truth. However, such commentary can be expressed in a variety of ways. Since the evaluation metrics used in existing video captioning tasks, such as BLUE, focus mainly on the presence or absence of expected words, we are concerned about whether the emotional and descriptive commentary generation can be evaluated appropriately. There is room to consider appropriate evaluation metrics for the novel dataset.

There are some typos such as fist -> first in line 450. The authors should check and correct typos throughout the paper.

**Suitability:**

3

---

### Official Review · Reviewer_xRYR · 2024-05-26

**Rating:** 4
**Confidence:** 4

**Summary:**

The paper discusses the challenge of generating detailed video descriptions that incorporate domain-specific information, specifically in the context of sports commentary generation. The authors introduce a novel dataset called BH-Commentary, consisting of basketball highlight videos with ground-truth commentaries from professional commentators. They also propose an end-to-end framework as a benchmark for basketball highlight commentary generation, which includes a prompt strategy to enhance alignment fusion among visual and textual features. Experimental results on the BH-Commentary dataset validate its usefulness and demonstrate the effectiveness of the proposed benchmark for sports highlight commentary generation.

**Strengths:**

The strengths of the paper include:

Novel dataset: The introduction of the Basketball Highlight Commentary (BH-Commentary) dataset is a significant strength. This dataset provides a valuable resource for researchers and practitioners in the field of sports commentary generation, as it consists of approximately 4K basketball highlight videos with ground-truth commentaries from professional commentators. The availability of such a dataset allows for more accurate evaluation and comparison of different approaches in this domain.

End-to-end framework: The proposed end-to-end framework for basketball highlight commentary generation is another strength of the paper. This framework offers a comprehensive solution to the challenge of generating detailed and domain-specific video descriptions. By designing a prompt strategy to enhance alignment fusion between visual and textual features, the framework aims to improve the overall effectiveness of the commentary generation process.

Experimental validation: The paper provides extensive experimental results on the BH-Commentary dataset, which demonstrate the validity of the dataset and the effectiveness of the proposed benchmark for sports highlight commentary generation. This empirical evidence adds credibility to the research findings and highlights the practical value of the proposed approach.

**Limitations:**

1. Anonymous websites load very slowly, which seriously affects the review effect.
2. The pipeline figure can be further optimized. The current layout of the entire image is not very beautiful.
3. In the introduction, authors need to spend more energy introducing their strengths and advantages. This allows the work to be highlighted more.

**Suitability:**

3

---

### Meta-Review · Area_Chair_sC4r · 2024-07-03

**Recommendation:** Accept (Poster)
**Confidence:** 5

**Metareview:**

The reviewers tend to recommend acceptance and the authors addressed some of the raised questions. I think this would be a valuable contribution as poster given also the dataset resources proposed by the paper.